# Dual Properties of Polyvinyl Alcohol-Based Magnetorheological Plastomer with Different Ratio of DMSO/Water

**DOI:** 10.3390/s21227758

**Published:** 2021-11-22

**Authors:** Norhiwani Mohd Hapipi, Saiful Amri Mazlan, Ubaidillah Ubaidillah, Siti Aishah Abdul Aziz, Seung-Bok Choi, Nur Azmah Nordin, Nurhazimah Nazmi, Zhengbin Pang, Shahir Mohd Yusuf

**Affiliations:** 1Engineering Materials and Structures (eMast) iKohza, Malaysian-Japan International Institute of Technology, Universiti Teknologi Malaysia, Jalan Sultan Yahya Petra, Kuala Lumpur 54100, Malaysia; hiwani87@gmail.com (N.M.H.); aishah118@gmail.com (S.A.A.A.); nurazmah.nordin@utm.my (N.A.N.); nurhazimah@utm.my (N.N.); bin.p@graduate.utm.my (Z.P.); shahiryasin@utm.my (S.M.Y.); 2Mechanical Engineering Department, Faculty of Engineering, Universitas Sebelas Maret, Jl. Ir. Sutami 36A Kentingan Jebres, Surakarta 57126, Indonesia; 3Department of Mechanical Engineering, The State University of New York, Korea (SUNY Korea), 119 Songdo Moonhwa-Ro, Yeonsu-Gu, Incheon 21985, Korea; 4Department of Mechanical Engineering, Industrial University of Ho Chi Minh City (IUH), 12 Nguyen Van Bao Street, Go Vap District, Ho Chi Minh City 70000, Vietnam

**Keywords:** hydrogel, magnetorheological plastomer, soft sensors, skin electronic, polyvinyl alcohol

## Abstract

Polyvinyl alcohol (PVA)-based magnetorheological plastomer (MRP) possesses excellent magnetically dependent mechanical properties such as the magnetorheological effect (MR effect) when exposed to an external magnetic field. PVA-based MRP also shows a shear stiffening (ST) effect, which is very beneficial in fabricating pressure sensor. Thus, it can automatically respond to external stimuli such as shear force without the magnetic field. The dual properties of PVA-based MRP mainly on the ST and MR effect are rarely reported. Therefore, this work empirically investigates the dual properties of this smart material under the influence of different solvent compositions (20:80, 40:60, 60:40, and 80:20) by varying the ratios of binary solvent mixture (dimethyl sulfoxide (DMSO) to water). Upon applying a shear stress with excitation frequencies from 0.01 to 10 Hz, the storage modulus (G′) for PVA-based MRP with DMSO to water ratio of 20:40 increases from 6.62 × 10^−5^ to 0.035 MPa. This result demonstrates an excellent ST effect with the relative shear stiffening effect (RSTE) up to 52,827%. In addition, both the ST and MR effect show a downward trend with increasing DMSO content to water. Notably, the physical state of hydrogel MRP could be changed with different solvent ratios either in the liquid-like or solid-like state. On the other hand, a transient stepwise experiment showed that the solvent’s composition had a positive effect on the arrangement of CIPs within the matrix as a function of the external magnetic field. Therefore, the solvent ratio (DMSO/water) can influence both ST and MR effects of hydrogel MRP, which need to be emphasized in the fabrication of hydrogel MRP for appropriate applications primarily with soft sensors and actuators for dynamic motion control.

## 1. Introduction

Sensors are small machines or devices purposely employed to detect changes in their environment by changing the information received from other electronics, usually a computer processor. Researchers have developed a number of approaches to improve the reliability of sensor systems, while not increasing the cost of production. In accordance, soft sensors have recently gained growing interest due to their advanced technology that can sense a small-scale change with high accuracy [1,2] For example, soft sensors have been widely used in electronic devices such as wearable skin-electronic devices to detect and predict human motion, thought and activities [3,4,5,6]. In wearable skin-electronic devices, a pressure signal is frequently utilized to relay human action to the device [7]. Therefore, the materials used to fabricate this sensor must be soft, flexible, stretchable, and human-friendly with high sensitivity and flexibility. Generally, soft materials such as hydrogel have been widely used in the fabrication of wearable strain or piezo resistance sensors due to their soft, hydrophilic nature, high stretchability and biocompatibility [8,9,10].

Recently, soft-solid materials, known as PVA-based MRPs, are the newest type of MR material that have been developed by incorporating micron-sized magnetic particles such as carbonyl iron particles (CIPs) into a low cross-linking polymer matrix such as hydrogel [11,12]. Due to the low cross-linking density of the hydrogel matrix, the CIPs are highly mobile in the polymeric matrix resulting in a more significant MR effect than solid-based MR materials [12]. Initially, the suspended CIPs in the PVA-based MRP are randomly distributed within the polymer matrix before rearranging into chain-like structures driven by the magnetic force. The fabrication of a PVA-based MRP is a one-step process involving mixing the CIPs into a conventional PVA hydrogel. Conventional PVA hydrogels are hydrophilic three-dimensional polymeric materials swollen by 80–90 wt.% of water, and are the most promising candidates for soft actuators and sensors [13]. Moreover, the biocompatibility, flexibility and stability of PVA hydrogels under several environmental conditions allows the hydrogel to mimic human soft tissues in the fabrication of strain/pressure sensors to detect human motion [14].

To date, a growing number of hydrogel sensing devices have been developed with various assembly principles [15,16,17]. For instance, Cai et al. [15] developed highly stretchable hydrogel strain sensors that can sustain severe elastic deformation (up to 1000%) with a high gauge factor of 1.51 using a conventional PVA hydrogel. Due to its hydrophilic behaviour, the PVA hydrogel can be chemically and physically cross-linked [11,18]. A chemically cross-linked PVA hydrogel has been developed using borax as a crosslinker agent that exhibited the shear stiffening (ST) phenomenon due to the existence of borax-oxygen weak interactions [19,20]. Additionally, Zhao et al. [21] suggested the materials that possess shear stiffening (ST) behaviour are excellent candidates in designing sensors due to their flexibility and conductivity. For example, PVA-based MRP exhibits ST behaviour, and can resist impact loading which is beneficial for use in piezoresistive-type sensors [22]. ST is a common phenomenon in which the viscosity increases when the materials are exposed to external stress beyond their critical shear rate [23]. Consequently, ST materials can passively react to external applied forces without any stimulus such as a magnetic field, allowing the material to respond with less power consumption [24]. Typically, materials that possess ST properties are softer under normal condition and become more rigid in response to external stresses such as high impact or extrusion. Other than that, no external power source or stimuli are required for inception by utilizing the ST properties. Zhao et al. [21] suggested that the materials that possess ST behaviour are excellent candidates for designing sensors due to their flexibility and conductivity. For example, materials that exhibit ST behaviour can resist impact loading, which is beneficial in piezoresistive-type and pressure strain sensors [10,22]. Thus, by having dual properties of the ST and MR effect, a PVA-based MRP can be potentially be used in the fabrication of sensors, especially as pressure or strain sensors, as illustrated in Figure 1.

A number of studies on MR materials have reported and highlighted both ST and MR effect characteristics by studying factors that influence their performance. For example, Wang et al. [23] studied a novel multifunctional polymer composite made from a polyurethane (PU) based MRP, which exhibited excellent ST behaviour and the MR effect. The obtained properties were mainly related to cross-linking bonds of the polymer matrix and the formation of CIPs in chain structures induced by the magnetic field. On the other hand, Liu et al. [25] studied the influence of cross-linked bonds of MR gel and the concentration of solid particles (CIPs) on the ST behaviour of MR gel. The relative shear stiffening effect (RSTE) was used to evaluate the ST behaviour of the material. The result demonstrated that the RSTE was highly influenced by the number of cross-linked bonds of the polymer matrix and the CIPs content. Moreover, Wang et al. [26] found that the ST performance could be correlated to the movement of CIPs within the polymeric matrix, which was essential to understand the CIPs’ behaviour inside the matrix phase. To further understand the relationship between the ST behaviour and CIPs mobility within the polymeric matrix, a transient stepwise experiment on the MR material can be carried out. For example, An et al. [27] performed a transient stepwise experiment with a tri-block copolymer MR gel to prove that the particles were easier to rearrange in the gel matrix. The results showed that the MR gel displayed strong stiffening properties under the external magnetic field, where the storage modulus increased up to 6000%. This higher storage modulus of MR gel was due to greater mobility and strong rearrangement of the particle network in the grease medium with an applied magnetic field.

According to Gupta et al. [28], the mechanical properties of the PVA hydrogel itself, such as toughness and tensile strength, were significantly affected by solvent composition. Thus, it is expected that varying the compositions of solvent during the preparation of PVA-based MRP would impact both ST and MR performances of the PVA-based MRP. Nevertheless, a study related to the influence of solvent compositions on both ST and MR performance, mainly on the PVA-based MRP, has been rarely documented. In fact, most studies related to PVA-based MRP were only focused on the MR effect performance. In contrast, a study on the respective ST behaviour is lacking, even though information about ST behaviour is essential as a future reference in determining related potential applications of PVA-based MRP. Before examining the electrical characteristics further, it is essential to analyse dynamic viscoelastic properties using an oscillatory shear rheometer. Thus, in this work, a group of PVA-based MRPs was synthesized in varying concentrations of binary solvent (dimethyl sulfoxide (DMSO) to water), and the dual properties of ST behaviour and MR performance under a dynamic oscillatory testing mode were conducted. Moreover, a transient stepwise experiment was performed to investigate the relationship between the rearrangement of the magnetic particle network under magnetic fields with ST behaviour. Lastly, a mechanism is proposed to further understand the movement of CIPs within the polymeric matrix upon application of the magnetic field.

## 2. Materials and Methods

PVA with a ≥98% degree of hydrolysis and an average molecular weight of 60,000 g/mol (Merck Company, Germany) was used to prepare the precursor solution. Sodium tetraborate decahydrate (borax), 20 Mule Team BoraxTM (drug store) was used as a cross-linking agent. DMSO brand ChemAR was supplied by Systerm Chemicals and deionized water was used as a solvent. These three chemicals were used to prepare a matrix-based PVA-based MRP. On the other hand, CIPs (CC type) with a size of ~3 µm were used as magnetic particles purchased from the BASF company. This CC type of CIP has a magnetic saturation, Ms, and a retentivity value, Mr, of 137.06 emu/g and 302.42 × 10^−3^ emu/g, respectively.

First, PVA-based MRP samples were prepared with 7.5% (*w*/*v*) PVA solution by diluting 8.1 g of PVA beads into binary mixtures of DMSO to water with various weights of DMSO:water ratio (20:80, 40:60, 60:40 and 80:20) at 80 °C for 2 h. A magnetic bar was used for continuous gentle stirring to ensure the homogeneity of the PVA solution. Next, the solution was cooled at room temperature before CIPs with 70 wt.% were added to each solution. Thus, four types of products were obtained as HMRP-20, HMRP-40, HMRP-60 and HMRP-80 for ratios of DMSO to water of 20:40, 40:60, 60:80 and 80:20, respectively. The detailed compositions of the mass fraction ratio of CIPs to the matrix for all samples are shown in Table 1. Then, the mixture was thoroughly mixed using mechanical stirring for 10 min before the cross-linking agent (borax solution) was added. Prior to this process, a borax solution (3% *w*/*v*) was prepared by dissolving the borax powder in deionized water. After the addition of the cross-linking agent, the as-prepared PVA-based MRP was mixed thoroughly using a mechanical stirrer. Next, the obtained PVA-based MRP was preserved overnight at ambient temperature before testing.

Environmental scanning electron microscopy (ESEM, 20 kV) with energy dispersive system analysis (EDS) was carried out using a JEOL JSM-6701 to obtain images of CIP distribution inside the PVA-based MRP matrix. The PVA-based MRP samples were first coated with platinum by an ion sputtering method before the ESEM analysis to avoid a charging effect that could cause blurry images. The rheological properties of the hydrogel MRP samples were determined using a commercial rheometer (Model: MCR 302 Anton Paar, Austria) equipped with an external magneto-controllable accessory, MRD 70/1T. The sample was placed in a parallel plate of 20 mm in diameter with a gap of 1 mm in thickness. Two types of measurements were conducted: a shear frequency sweep test and a magnetic-induced test to study the shear stiffening and MR performance of hydrogel MRP, respectively. The shear frequency sweep test was performed at excitation frequencies ranging from 0.01 to 10 Hz with a constant strain of 0.01%. Meanwhile, the current sweep test was conducted under a linear excitation magnetic field of 0 to 800 mT with a constant strain of 0.01% and a frequency of 1 Hz. Additionally, the transient response test under a stepwise magnetic field was conducted at a constant external magnetic field of 500 mT. From the transient response test, the dimensionless storage moduli of the PVA-based MRP with different solvent ratios were calculated and analysed. All the rheology testing were conducted at a fixed room temperature.

## 3. Results and Discussion

### 3.1. Microstructural Analysis

Micrograph observation was conducted in the absence of a magnetic field to investigate the distribution of CIPs within the matrix. Figure 2 shows the microstructural observation and EDS mapping analysis of the PVA-based MRP sample. As shown in Figure 2a, the CIPs were observed to be homogeneously distributed in the matrix of the PVA-based MRP, suggesting a uniform distribution of CIPs inside the matrix. Moreover, Figure 2b indicates the existence of a transparent layer by showing a PVA layer that enwrapped the CIPs, which is consistent with the findings by Xu et al. [11]. Meanwhile, in Figure 2c, the EDS analysis shows the presence of three elements in the PVA-based MRP sample, which are carbon (C), oxygen (O) and iron (Fe). The two elements, C and O, represented the PVA structure, whilst Fe represented iron for the CIPs. As the PVA (C_2_H_4_O) itself contains a hydrogen (H) bond, the element should be presented in the EDS analysis; however, it could not be detected due to limitation of the EDS detector. According to Wilson et al. [29], the existence of hydrogen bonding in the polymeric matrix would increase the stability of the CIPs over time, and simultaneously reduce the settling issue.

### 3.2. Shear Stiffening Effect of Hydrogel MRP

The PVA-based MRP displayed an ST effect whenever the storage modulus, G′ of the material changed with a change of frequency without applying an external magnetic field. Figure 3 shows the ST effect of PVA-based MRP samples with different solvent ratios, without the magnetic field (off-state). The frequency was applied from 0.01 to 10 Hz with a constant strain amplitude of 0.01%. In order to compute the ST effect of the PVA-based MRP at different ratios of DMSO: water, the absolute ST effect (ASTE) and relative ST effect (RSTE) were used, as shown in Equations (1) and (2).
(1)$$\mathrm{ASTE}={\;G{}^{\prime }}_{\max}-{G{}^{\prime }}_{\min},$$
(2)$$\mathrm{RSTE}\%=\frac{{G{}^{\prime }}_{\max}-{\;G{}^{\prime }}_{\min}}{{G{}^{\prime }}_{\min}}\times 100\%$$
where, G′_max_ is the maximum storage modulus of the PVA-based MRP induced by the shear frequency at *f* = 10 Hz, and G′_min_ is the initial storage modulus of PVA-based MRP at *f =* 0 Hz.

In the absence of an external magnetic field, the storage modulus, G′ of all samples increased with increasing shear frequency from 0.01 to 10 Hz, suggesting a change of ST behavior in the PVA-based MRPs. The increase of storage modulus reached saturation at high shear frequency. Moreover, it was noticed that t G′ was higher than G″ for all samples at higher frequency, which indicated stiffer samples. The result suggests that the materials transformed to a harder state in response to the external shear stress in terms of shear frequency, which signifies the ST effect phenomenon [30]. Prior to the change of ST effect in the PVA-based MRP samples, it was important to determine the physical state of the PVA-based MRP to distinguish the states of liquid-like and solid-like PVA-based MRP by analyzing variation of G′ and G″ as a function of frequency. Generally, the material would exhibit liquid-like (viscous) behavior when G″ > G′. In contrast, when the G′ > G″ the PVA-based MRP was in a solid-like (elastic) state. For example, as shown in Figure 3a for HMRP-20, it was noticed that the loss modulus, G″, is greater than the storage modulus, G′ until reaching the “cross-over” (CR) point at a frequency of ~3.5 Hz. This CR point suggested that at the beginning of the reaction, the HMRP-20 behaved in a liquid-like state, which was viscous. However, as the frequency increased, the material became stiffer above the CR point and exhibited a solid-like state.

In fact, all PVA-based MRP samples possessed similar behavior, acting in a liquid-like state at the beginning stage, then progressively changing to solid-like behavior at higher applied frequencies, particularly above the CR point for each sample, as G′ > G″. Moreover, it was noted that the CR point for the PVA-based MRP samples shifted to the left in correspondence with the increase in DMSO content. In fact, the CR point for the HMRP-40 was at about ~1.39 Hz, followed by HMRP-60 and HMRP-80 at about ≤0.70 Hz, showing a shorter range of the transition from the liquid-like to a solid-like state of HMRP. Thus, HMRP-60 and HMRP-80 were said to dominantly exhibit elastic behavior at very low-frequencies. These results imply the influences and changes of solvent content on the physical and rheological properties of the PVA-based MRPs. Hence, the samples could be separated into two phases based on their physical states, which were liquid-like PVA-based MRP due to water-rich (HMRP-20 and HMRP-40) compositions, and solid-like PVA-based MRP due to DMSO-rich (HMRP-60 and HMRP-80) compositions, particularly at the off-state condition. These findings agree well with Wong et al. [31], who stated that solutions containing more than 50% of DMSO were referred to as low water concentration, or DMSO-rich, and vice versa.

Furthermore, with the addition of DMSO as a cosolvent in the PVA-based MRP, the gel networks were strengthened, resulting in an increase of the initial storage modulus, G′_0_ of the PVA-based MRP. In the presence of DMSO as an organic solvent in the PVA-based MRP, the concentration of boron species that bound with the vinyl alcohol groups in the PVA matrix was improved due to the complexation of borate ions with the PVA chains. In fact, according to Riedo et al. [32], the presence of cosolvent solutions such as DMSO had led to a less attractive environment for the borate ions, causing the ions to migrate to the PVA chains. As the number of cross-links between borate ions and the PVA chains increased, the complexation chains induced stiffer PVA-based MRP, thus increasing the storage modulus of the material. Based on the different trends observed in Figure 2, discussion of properties can be separated into two parts, mainly related to the liquid-like and solid-like behavior of PVA-based MRP, as depicted in Table 2. Moreover, the values of ASTE and RSTE for PVA-based MRP samples were calculated and are displayed in Table 2 to show the correlation of the influence of DMSO content on ST behavior.

Table 2 illustrates that for the water-rich PVA-based MRP, such as HMRP-20 and HMRP-40, the ASTE value decreased with an increase of DMSO content. A similar trend was observed for the DMSO-rich PVA-based MRP, where the ASTE value also decreased with an increase of DMSO content. At the off-state condition, the ASTE value, which refers to the change of G′ (∆G), was manipulated by the ability of the CIPs to move within the polymeric matrix, responding to the external shear frequency. As the shear frequency increased, the mobility of CIPs was restricted, resulting in an increase in the G′ of the PVA-based MRP. This phenomenon was also related to the ST effect. In the absence of a magnetic field, a hydro cluster occurred due to the clustering of CIPs within the polymer matrix at certain shear frequencies. A hydro cluster is a phenomenon in which the particles are forced closer to each other by the external shear stress, forming a self-organized microstructure. The formation of the hydro cluster caused by the hydrodynamic forces restricts the motion of the CIPs in the matrix resulting in an increase of the RSTE values, which was displayed by HMRP-20 and HMRP-40, at 52,827% and 47,916%, respectively. For DMSO-rich samples, HMRP-60 and HMRP-80, the RSTE values were lower, at 3970% and 1673%, respectively. As the hydro cluster phenomenon coincided with the ST effect, where it involved the mobility of CIPs within the polymer matrix, it could be said that the water-rich samples were expected to exhibit a large ST effect compared to DMSO-rich samples due to the ease of CIP mobility. On the other hand, for DMSO-rich samples, when the DMSO content were increased the stiffness of the PVA-based MRP itself increased and caused a difficult internal movement for CIP clustering, which resulted in a decrease in the RSTE value. This observation agrees well with the finding by Wang et al. [33].

However, under an external magnetic field, the ST effect was manipulated mainly by CIP movement and formation of chain-like structures along the magnetic flux direction. Therefore, to further evaluate the inner performance of PVA-based MRP samples, the synergistic effect of both magnetic field strength and shear frequency on the rheological performances of PVA-based MRP was investigated. Figure 4 depicts the ST properties of different DMSO contents of PVA-based MRPs, and shear frequency in the presence of a magnetic field of 180 mT.

As shown in Figure 4, it was observed that the G′ of all PVA-based MRP samples increased with increasing shear frequency from 0.01 to 10 Hz under the magnetic field, which proved that the PVA-based MRP would become stiffer at a higher external shear frequency. In fact, the G′ was always greater than the G″, which means that all samples exhibited solid-like behaviour when exposed to the magnetic field due to the strengthening of CIP interactions [34,35] Moreover, it was also proved that the PVA-based MRP exhibited a synergistic effect as the PVA-based MRP responded to two different external stimuli simultaneously, namely shear frequency and magnetic field. In the condition under the magnetic field, the CIPs played a big role in strengthening the PVA-based MRP by rearranging themselves into chain-like structures depending on the strength of the magnetic field. The HMRP-20 sample exhibited the highest G′_max_, at approximately 1.708 MPa under a magnetic field of 180 mT. The G′_max_ showed a decreasing trend with an increasing DMSO content, which was 0.922 MPa for HMRP-40, followed by HMRP-60 and HMRP-80, which were 0.248 and 0.197 MPa, respectively. The decrement of G′_max_ could be explained by the movement of the CIPs within the polymer matrix as a result of magnetic field excitation during shear formation. For instance, the CIPs were more likely to move and form complex chain-like structures due to magnetic field excitement in the water-rich PVA-based MRP compared to the DMSO-rich PVA-based MRP. On the other hand, for DMSO-rich samples, the movement of CIPs was greatly hindered by the polymer matrix molecular chains due to increased matrix stiffness. Furthermore, under the magnetic field, the ST behavior of PVA-based MRP samples was affected by the magnetic particles like CIPs within the polymer matrix. During shear under an external magnetic field, the CIPs were magnetized, started to move and formed chain-like structures that restricted the movement of PVA chains in the shear direction.

The transient response of PVA-based MRP samples to a stepwise magnetic field was tested to further understand the effect of DMSO content on the structure’s formation process driven by the external magnetic field. Figure 5 shows the time-rheological response of a stepwise magnetic field of PVA-based MRP samples. In this experiment, the storage modulus of PVA-based MRP samples was measured upon applying a stepwise magnetic field, where a magnetic field of 500 mT was instantaneously applied to and removed from the sample at 120 and 240 s, respectively. Moreover, the rearrangement process of CIPs in the PVA-based MRP matrix was further investigated by analyzing G′s time evolution in response to an instantaneously applied magnetic field. Thus, Figure 5b shows the dimensionless transient responses of the PVA-based MRP samples. The dimensionless storage modulus can be calculated using Equation (3) [36];
(3)$$\mathrm{Dimensionless},\;G{}^{\prime }=\frac{\;G{}^{\prime }}{{\;G{}^{\prime }}_{\max}}$$
where, G′_max_ is the maximum value of G′ for the PVA-based MRP samples in the range of 120–240 s.

Based on Figure 5, it can be deduced that the solvent ratio between DMSO and water had a major impact on the dynamic property of PVA-based MRP samples. As shown in Figure 5a, the PVA-based MRP with less DMSO content (HMRP-20) had a larger G′ up to 1.78 MPa. The G′ decreased with an increasing DMSO content, and was 1.65, 1.62 and 1.33 MPa for HMRP-40, HMRP-60 and HMRP-80, respectively. When exposed to an external magnetic field at the time range of 120–240 s, the G′ of PVA-based MRP samples increased sharply and suddenly disappeared after the magnetic field was removed after 240 s. Furthermore, G′ showed an increasing trend under the magnetic field with decreasing DMSO content, which means that the CIPs had a stronger formation of particle chains in the PVA-based MRP with the lower DMSO content. Moreover, Figure 5b shows the dimensionless transient response for each PVA-based MRP sample at 120 to 240 s to compare the changing time for the CIPs. Under a sudden application of a magnetic field of 500 mT, the distribution of CIPs shifted from a random orientation to a steady chain-like structure in the PVA-based MRP matrix. According to Xu et al. [37], when the value was beyond 0.99, the dimensionless G′ would fluctuate between 0.99 and 1 ascribed to the breaking and reformation of particle chain-like structures. Therefore, the time for the CIPs to achieve a dimensionless G′ of 0.99 could be considered the most stable formation of CIPs chain-like structures. The interaction of CIPs within the PVA-based MRP upon the magnetic field applications is illustrated in Figure 6.

At first, the CIPs were randomly distributed within the matrix without the application of an external magnetic field (off-state condition). When a magnetic field was applied to the samples, the CIPs magnetized and appeared to form chain-like structures in the direction of the magnetic field. For water-rich samples, the CIPs easily formed chain-like structures due to less chemical cross-linking between the PVA chains. On the other hand, for DMSO-rich samples, the formation of CIP chain-like structures was obstructed by the PVA chains due to more chemical cross-linking between PVA chains. As a result, the response of CIPs towards the magnetic field was more pronounced in PVA-based MRP samples containing less DMSO, such as HMRP-20 and HMRP-40, due to less intermolecular H-bonding that disrupted the movement of CIPs to form chain-like structures [28]. Moreover, the interaction of borate ions was greater in the DMSO-rich solvent than in the water-rich samples. Thus, as displayed in the inset of Figure 4b, it can be assumed that the movement of CIPs was easier in PVA-based MRP sample with lower DMSO content as the CIPs encountered less resistance from the polymer matrix with a lower viscosity (water-rich samples). For example, sample HMRP-20 required a shorter time to form stable chains, which was at 98 s, compared to other samples. In addition, the longest time for the PVA-based MRP sample to reach a dimensionless G′ of 0.99 was at 118 s by HMRP-80. The response time of the HMRP-80 was faster than the solid-like polyurethane MR gel (categorized as MRP) reported from a previous study, which took about 275 s to form stable chain-like structures [37].

### 3.3. Relative MR Effect and Damping Properties

The MR effect is an important characteristic to study the response of MR materials under the excitation of an external magnetic field [38]. Therefore, to further investigate the MR performance of PVA-based MRP samples, a magnetic field sweep test was conducted, and the MR effect for each sample was calculated using Equations (4) and (5):(4)$$\mathrm{Absolute}\;\mathrm{MR}\;\mathrm{effect}={\;G{}^{\prime }}_{\max}-{G{}^{\prime }}_{\min},$$
(5)$$\mathrm{Relative}\;\mathrm{MR}\;\mathrm{effect}\%=\frac{{G{}^{\prime }}_{\max}-{\;G{}^{\prime }}_{\min}}{{G{}^{\prime }}_{\min}}\times 100\%$$

The shear storage modulus, G′, of PVA-based MRP samples with different DMSO contents under an excitation magnetic flux density up to 800 mT, is shown in Figure 7a, whilst the absolute and relative MR effect are calculated and displayed in Figure 7b.

As illustrated in Figure 7a, G′ increased sharply along with the excitation magnetic field for all the PVA-based MRP samples. HMRP-20 exhibited the highest value of G′_max_, which was 2.13 MPa at a magnetic field of 800 mT, compared to others. The sample with the lowest DMSO content demonstrated the highest absolute (ΔG′) and relative MR effect as shown in Figure 7b. In general, the maximum shear modulus typically influenced the absolute MR effect, while the relative MR effect was affected by the initial shear modulus. Thus, the absolute and relative MR effects in this experiment showed a descending trend from 2.13 MPa for HMRP-20 to 1.83 MPa for HMRP-80. Furthermore, the relative MR effect also displayed a decreasing trend with increasing DMSO content. For example, the relative MR effects for HMRP-20, HMRP-40, HMRP-60 and HMRP-80 were 27,087, 18,231, 15,931 and 11,656%, respectively. As both absolute and relative MR effects resulted from restructuring of CIPs under the magnetic field to form chain-like structures, this result can used to denote the movement and response of CIPs in different solvent concentrations. Thus, PVA-based MRP sample with the highest DMSO content created a stronger intermolecular interaction between the DMSO and water, limiting the movement of CIPs as a reaction to the magnetic field. Meanwhile, an increase of DMSO content led to a higher initial storage modulus that caused a reduction in the relative MR effect. Figure 8 displays the loss factor denoted as Tan δ to evaluate the damping properties of PVA-based MRP samples. The damping property is the measure of how well the materials can get rid of energy in terms of heat loss.

From Figure 8, the loss factor displays a decreasing trend with increasing DMSO content. For example, it was noted that the initial loss factor for a water-rich PVA-based MRP sample with 20 wt.% DMSO (HMRP-20) was 0.711, which was 3.09 times larger than the DMSO-rich sample with 80 wt.% DMSO content. Besides, the strength of the magnetic flux density affected the movement of the CIPs and caused changes in the damping properties. At a lower magnetic field, 0.4 T, the loss factor for HRMP showed a decreasing trend with increased DMSO content. However, as soon as the magnetic flux density exceeded 0.4 T, the loss factor acted oppositely. The reason was that, as at the initial excitation of the magnetic field, the CIPs could move freely by changing their positions following the direction of the magnetic field. At this time, the damping properties (energy dissipation) were primarily caused by interfacial slipping between the CIPs and the matrix. That was why the water-rich samples (HMRP-20 and HMRP-40) had a higher loss factor at the earlier application of the magnetic field, as movement of CIPs induced a significant amount of dissipation energy. Nonetheless, in the case of DMSO-rich samples (HMRP-60 and HMRP-80), the loss factor was lower, denoted by the difficulty of the CIPs to move due to the restriction effect of the matrix [39].

Likewise, the loss factor leveled off with further increase of magnetic flux density above 0.4 T, indicating that the interactions between the CIPs had reached the saturated stage. This meant that at this stage the particles received maximum forces and were in a stable state resulting the value of the loss factor reaching a plateau. Moreover, as shown in Figure 7, HMRP-80 had a higher loss factor at a magnetic flux density above 0.4 T due to the matrix’s higher restriction, which reduced the interaction between the CIPs. According to Yang et al. [39], the damping properties of MR materials are derived from intrinsic damping (particle-matrix interaction) and interface damping (friction loss due to slip between particles and matrix). However, in the current study, as the content of CIPs was the same for all PVA-based MRP samples, the intrinsic damping could be neglected. Thus, the damping properties were mainly affected by interface damping resulting from the friction between the polymer matrix and the CIPs itself. So, it can be concluded that HMRP-80 samples exhibited a larger dissipation energy than others due to higher friction between the CIPs and the matrix during the CIPs’ movement. Interestingly, this behavior suggests that the physical state of HMRP-80 was more likely to be as a solid-like MR gel (MR plastomer). At the same time, the other samples showed characteristics of liquid-like MR gels [37].

## 4. Conclusions

In this study, a group of PVA-based MRPs was fabricated by using carbonyl iron particles (CIPs) and polyvinyl alcohol (PVA) as a polymer matrix with different ratios of dimethyl sulfoxide (DMSO) to water. The effect of different ratios of DMSO concentration on the ST and MR performances were investigated. The results showed that by changing the DMSO ratio, the physical states of PVA-based MRPs could change between liquid and solid-like conditions. Consequently, the changes affected the ST and MR performances. The ST and MR effects showed a decreasing trend with increasing DMSO content due to the polymer matrix’s restriction. The relative ST effect and MR effect of PVA-based MRPs with 20 wt.% DMSO (HMRP-20) were 52,827% and 27,087%, respectively. Moreover, the PVA-based MRP was physically and rheologically changed to a solid-like PVA hydrogel MRP when the DMSO content was 60 wt.% and above, which is a plasticine-like consistency (shape can be molded) compared to samples with lower DMSO content. Meanwhile, the result from transient responses proved that the CIPs moved more easily in the water-rich solvent than in the DMSO-rich solvent, as the formation of the polymer chain network was not as complex as in the DMSO-rich samples where the mobility of the CIPs was hindered. Furthermore, the PVA-based MRP exhibited a synergistic effect, where the ST and MR effects could be controlled at one time by altering the DMSO content in the PVA hydrogel MRP. Thus, these unique features confer advantages for the PVA hydrogel MRP to be used in broad applications, especially in dampers, sensors, actuators, and protection devices. More comprehensive study on the electrical conductivity on the PVA-based MRP should be conducted in the future for successful application of the proposed smart material as a potential sensor.

## Figures and Tables

**Figure 1 sensors-21-07758-f001:**
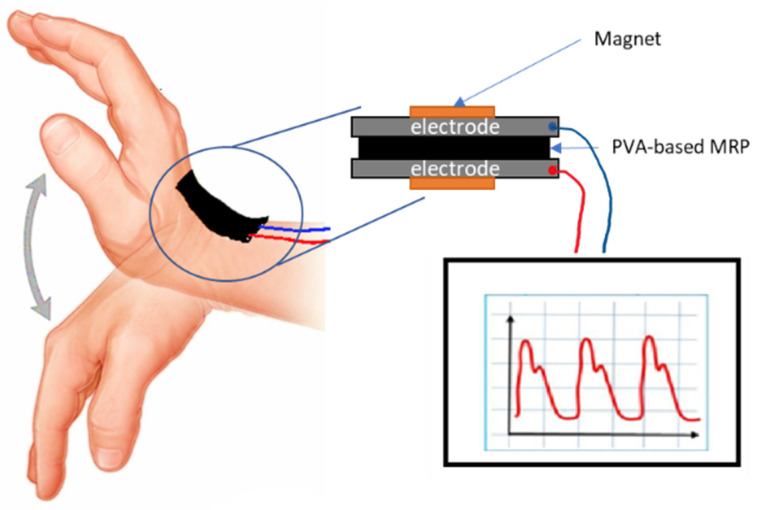
Basic structure of the proposed PVA-based MRP as a pressure/strain sensor to detect human motion (bending and stretching).

**Figure 2 sensors-21-07758-f002:**
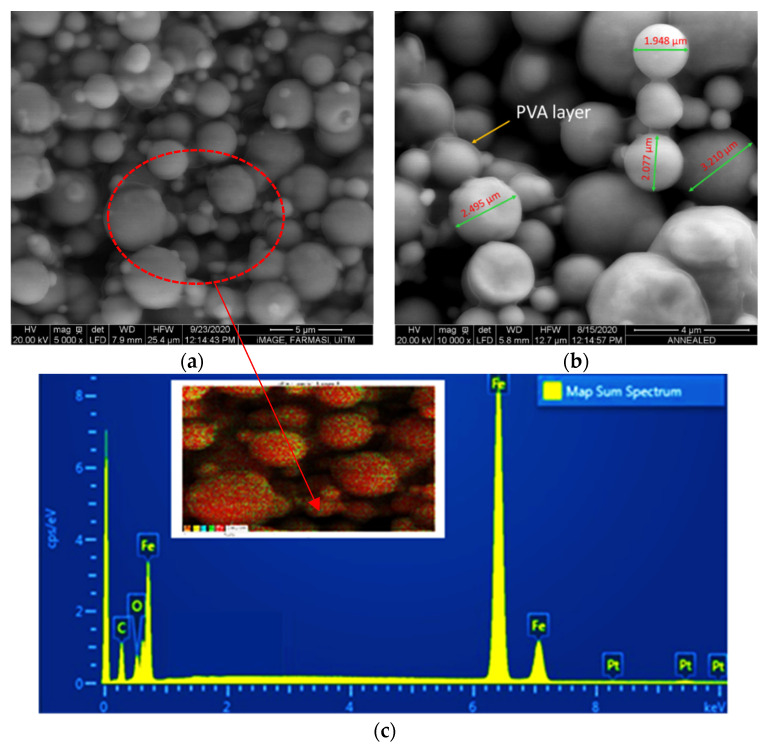
ESEM micrographs of the distribution of CIPs in the PVA-based MRP matrix, with magnifications of (**a**) ×5k and (**b**) ×10k, and (**c**) EDS mapping of the sample. Dot scanning analysis was conducted on the sample image as shown in the red dot circled in (**a**).

**Figure 3 sensors-21-07758-f003:**
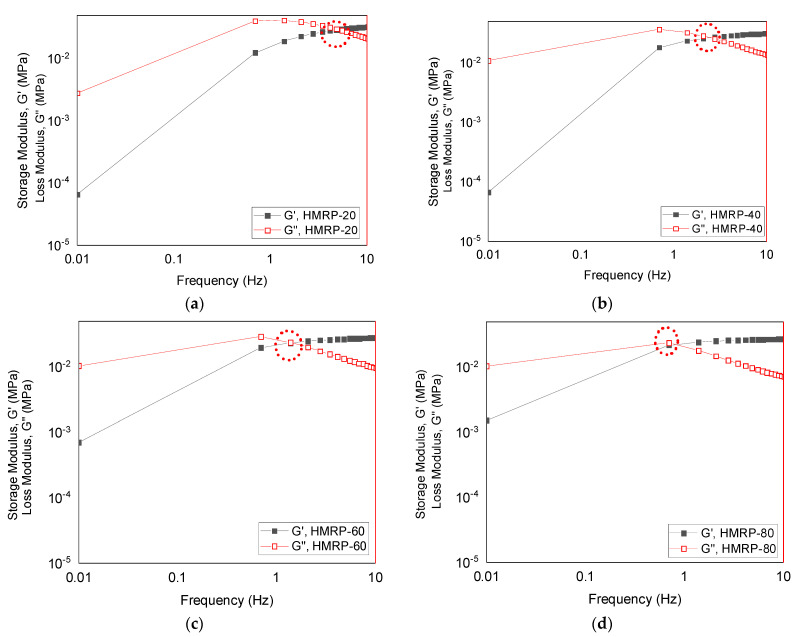
Shear storage and loss modulus of PVA-based MRP samples with different ratios of DMSO:water content with continuously changes in shear frequencies for (**a**) HMRP-20, (**b**) HMRP-40, (**c**) HMRP-60 and (**d**) HMRP-80.

**Figure 4 sensors-21-07758-f004:**
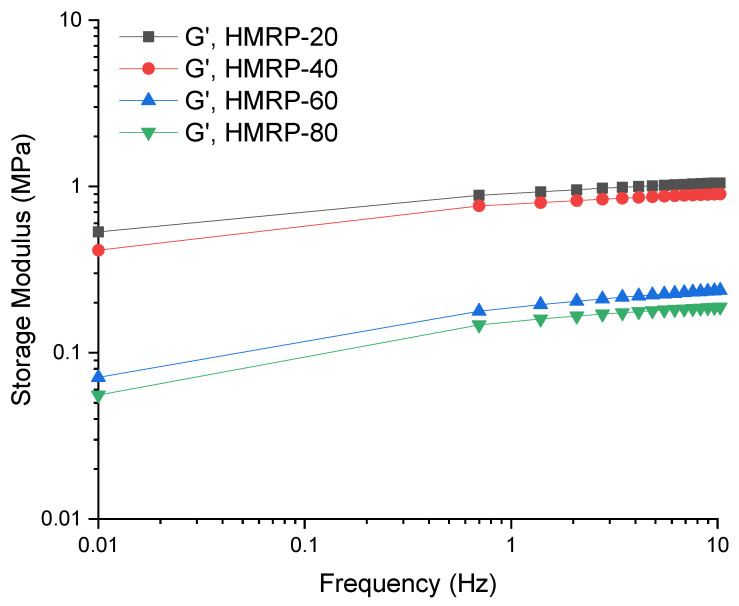
Shear storage modulus of PVA-based MRP samples with different ratios of DMSO:water under continuously changing frequency.

**Figure 5 sensors-21-07758-f005:**
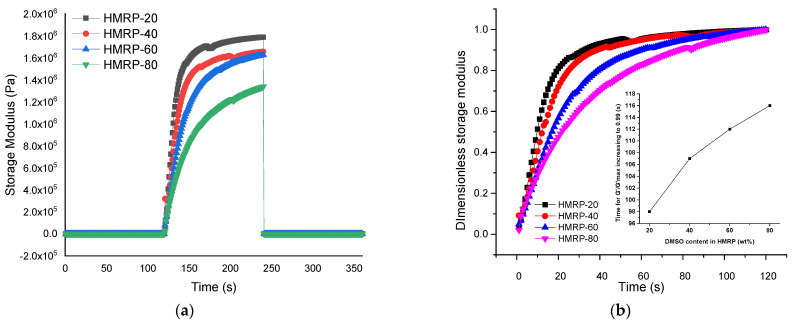
Time dependence of G′ for PVA-based MRP with different solvent ratios in response to a stepwise magnetic field (**a**) and the dimensionless transient response of G′ (**b**) under a magnetic field of 500 mT.

**Figure 6 sensors-21-07758-f006:**
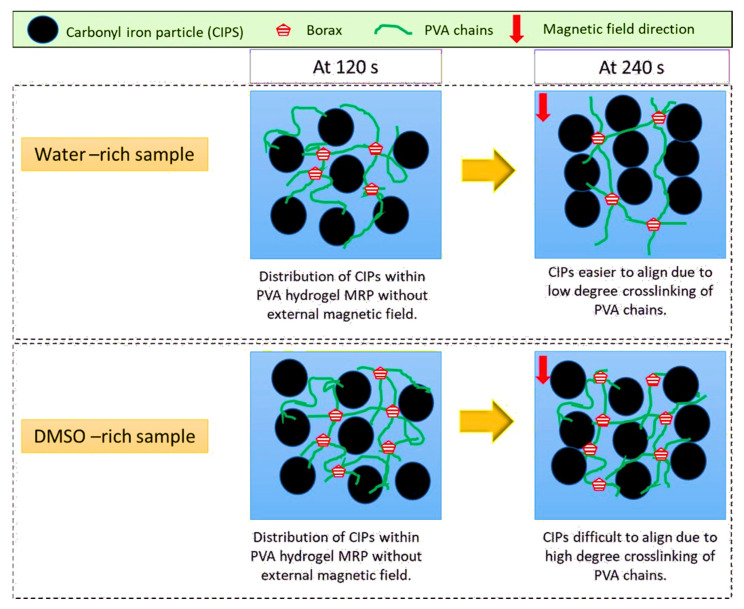
Schematic illustration of the mechanism of CIP movement upon the application of external magnetic field in water-rich and DMSO-rich PVA-based MRP.

**Figure 7 sensors-21-07758-f007:**
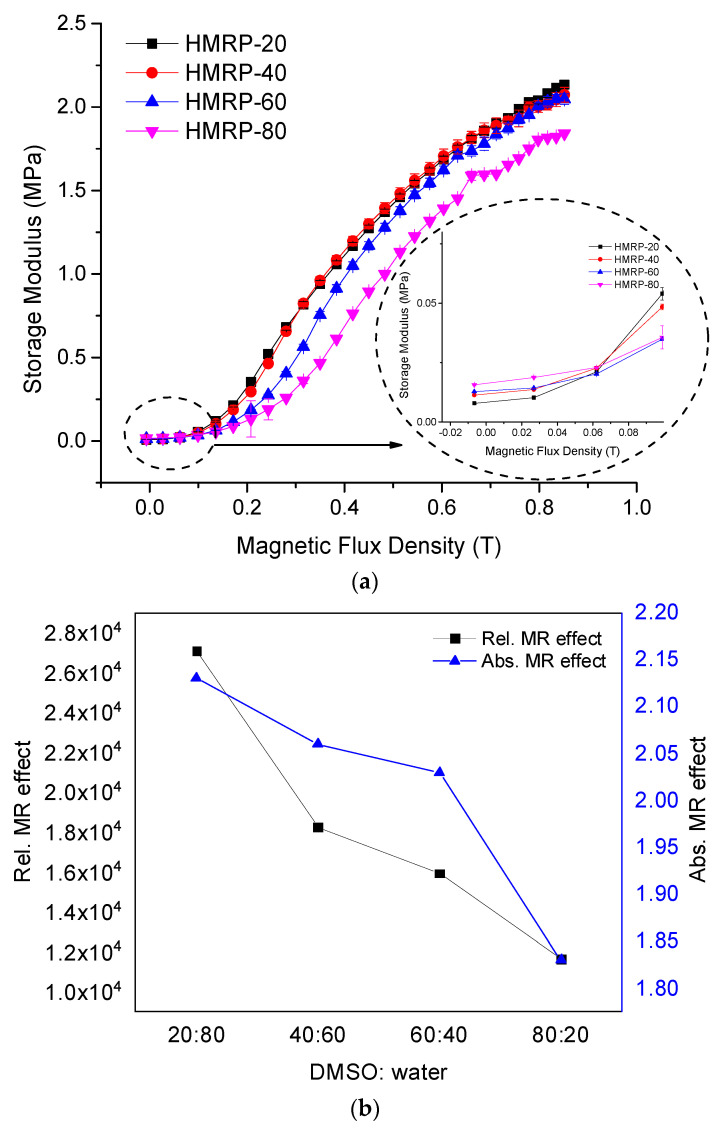
Shear storage modulus of PVA-based MRP samples with the different ratios of DMSO:water (**a**), and the calculated ASTE and RSTE (**b**) of the samples under a continuously changing magnetic flux density.

**Figure 8 sensors-21-07758-f008:**
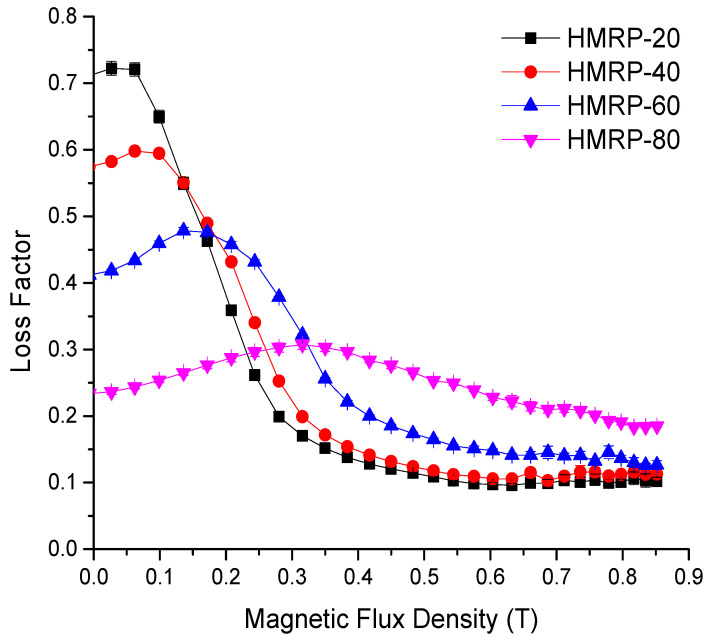
Relationship between loss factor and magnetic flux density for PVA-based MRP samples with different DMSO contents.

**Table 1 sensors-21-07758-t001:** Compositions of hydrogel MRP samples.

Samples	DMSO: Water Ratio [wt.%]	CIP [wt.%]
HMRP-20	20:80	70
HMRP-40	40:60	70
HMRP-60	60:40	70
HMRP-80	80:20	70

**Table 2 sensors-21-07758-t002:** G′_max_, G′_min_, ASTE, and RSTE of PVA-based MRP samples under shear frequency tests.

Sample	G′_max_/MPa	G′_min_/MPa	ASTE/MPa	RSTE/%
HMRP-20	0.035	6.62 × 10^−5^	0.035	52,827
HMRP-40	0.032	6.73 × 10^−5^	0.032	47,916
HMRP-60	0.028	7.06 × 10^−4^	0.028	3970
HMRP-80	0.027	0.001	0.026	1673

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
