# Peer review of "Dual Properties of Polyvinyl Alcohol-Based Magnetorheological Plastomer with Different Ratio of DMSO/Water"

_sensors, 2021, doi:10.3390/s21227758_

Round 1

Reviewer 1 Report

The authors investigate experimentally the shear stiffening effect (ST) and the magneto-rheological effect (MR) of a Polyvinyl alcohol magneto-rheological plastomer under the influence of different solvent compositions by varying ratios of binary solvent mixture to water. One of the interesting things they were able to show is that the ST and MR effect have a downward trend with increasing solvent mixture content to water.

Similar effects regarding downtrend of the MR effect were in reported in recent theoretical and numerical works in the following papers:

Nika, G., Vernescu, B. Multiscale modeling of magnetorheological suspensions, Z. Angew. Math. Phys. 71 (1) (2020) pp. 1--19. https://doi.org/10.1007/s00033-019-1238-4

Nika, G., Vernescu, B. Micro-geometry effects on the nonlinear effective yield strength response of magnetorheological fluids, book chapter in: P. Donato, M. Luna-Laynez (eds.), Emerging Problems in the Homogenization of Partial Differential Equations, SEMA SIMAI Springer Series 10, (2021) pp. 1--16. https://doi.org/10.1007/978-3-030-62030-1_1

Moreover, in the above two articles, the MR fluid was shown to exhibit a solid-like behavior, similar to the effect the authors' describe on pg. 9, lines 303-305.

The results reported, are interesting and I recommend the paper for publication. I would strongly recommend the authors make the following changes/additions to their article:

  1. The authors have to cite the above two works
  2. ``Sensors are a small machine or device that is purposely...'' change to ``Sensors are small machines or devices that are purposely...''
  3. ``its environment...'' change to ``their environment...''

Author Response

Dear Reviewer, I hereby attached our response. Thank you

Reviewer 2 Report

This article studies the shear stiffening and magneto-rheological effect of PVA-based MR plastomer. Some points need to be addressed before accepted.

Line 140: Please provide the magnetic properties of the CIP, such as Ms and Mr.

Line 160: How to obtain the images of liquid or fluid by using SEM? Please provide the detailed sample preparation procedure.

Line 169: During the frequency sweep test, what is the increasing type of frequency? Linearly or logarithmically? This is should be cleared. Similar comments to the magnetic field sweep test.

What is the temperature during the rheological test? Is it fixed or non-controlled? If non-controlled, the results of this study will be meaningless due to the change of magnetic properties of CIP with temperature.

Line 196: Apparently, the EDS map is obtained by a dot scanning. Please indicate which point is the analysis area in the SEM images.

Line 205: From the reviewer’s opinion, the calculation results of ASTE and RSTE are not reliable, since the unreliable values of G’min. At frequency close to zero, the G’ measurement is very sensitive. From Fig. 3 we can know that the frequency is scanned linearly from 1 to 20, recording 30 data points. There is a big difference between the first G’ and the second G’, while the frequency only changes in a very small scale. I am not sure whether you pre-sheared the sample or not after pouring it to the plate. As pouring the sample to the plate and the gap adjustment procedure have a significant influence on the initial state of the sample, and how do you make sure that these four samples have the similar reference state? To eliminate this influence, it is better to perform the frequency sweep test logarithmically. In this way, an average G’ can be obtained when the frequency is approaching to zero. I suggest to re-perform the rheological results. If you still use the old experimental results, then please delete all the quantitative analysis of shear stiffening and magneto-rheological effect, since all the calculations based on G’min are doubtful.

Line 265: The data in Table 2 provides evidences to my above comments. The small G’min is highly influenced by the rheometer sensitivity, and they are not reliable. The error range (e.g., ±0.003) in this column is absolutely not correct, they are not the real value.

Line 355-356: this sentence is not clear. Please rephrase.

Author Response

Dear Reviewer, I hereby submitted the response

Round 2

Reviewer 2 Report

The authors have made required changes according to the comments.